# Psychometric evaluation of the 'Attitudes and Beliefs about Cardiovascular Disease (ABCD) Risk Questionnaire' with validation of a previously untested 'Intentions and Beliefs around Smoking' subscale

Mark Bowyer  ,[1] Hamid Yimam Hassen  ,[2] Hilde Bastiaens,[2] Linda Gibson[1]

[1]Institute of Health and Allied Professions, School of Social Sciences, Nottingham Trent University, Nottingham, UK
[2]Family Medicine and Population Health, Faculty of Medicine and Health Services, University of Antwerp, Wilrijk, Belgium

**Correspondence to**
Mark Bowyer;
mark.bowyer@ntu.ac.uk

## ABSTRACT

**Objectives** To provide evidence of validity, reliability and generalisability of results obtained using the Attitudes and Beliefs about Cardiovascular Disease (ABCD) Risk Questionnaire with a sample of the English population surveyed within the 'SPICES' Horizon 2020 Project (Nottingham study site), and to specifically evaluate the psychometric and factor properties of an as-yet untested five-item subscale relating to smoking behaviours.

**Design and setting** Community and workplace-based cross-sectional study in Nottingham, UK.

**Participants** 466 English adults fitting inclusion criteria (aged 18+ years, without known history of cardiovascular disease, not pregnant, able to provide informed consent) participated in the study.

**Intervention** We revalidated the ABCD Questionnaire on a sample of the general population in Nottingham to confirm the psychometric properties. Furthermore, we introduced five items related to smoking, which were dropped in the original study due to inadequate valid samples.

Primary and secondary outcome measures
1. Psychometric and factor performance of untested five-item 'smoking behaviours' subscale.
2. Psychometric and factorial properties in combination with the remaining 18 items across 3 subscales.

**Results** Analyses of the data largely confirmed the validity, reliability and factor structure of the original ABCD Risk Questionnaire. Sufficient participants in our study provided data against additional five smoking-related items to confirm their validity as a subscale and to advocate for their inclusion in future applications of the scale. Exploratory factor analysis and confirmatory factor analysis calculations support some minor changes to the remaining subscales, which may further improve psychometric performance and therefore generalisability of the instrument.

**Conclusions** An amended version of the ABCD Risk Questionnaire would provide public health researchers and practitioners with a brief, easy-to-use, reliable and valid survey tool. The amended tool may assist public health practitioners and researchers to survey patient or public intentions and beliefs around three key areas

## STRENGTHS AND LIMITATIONS OF THIS STUDY

⇒ Large sample (n=466) of English adults from the Nottingham UK population.
⇒ Sufficient case data to validate additional subscale related to attitudes and intentions of smokers.
⇒ Criterion validity not explored.
⇒ Full assessment of the utility of Attitudes and Beliefs about Cardiovascular Disease Risk Questionnaire in health promotion and cardiovascular disease prevention was not explored; further studies may be required to position the tool in clinical and public health practice.
⇒ The planned pre/post-intervention measurement and analysis were not possible due to COVID-19 interruption of fieldwork.

of individually modifiable risk (physical activity, diet, smoking).

**Trial registration number** ISRCTN Registry (ISRCTN68334579).

## INTRODUCTION
### Scientific background and rationale
In the UK, cardiovascular disease (CVD) is responsible for over 130 000 deaths per annum.[1] CVD morbidity is also the biggest contributor to the inequalities in healthy life expectancy between members of the wealthiest neighbourhoods and the most deprived.[2] In 2009, the National Health Service (NHS) Health Check[3] was established and more recently (2019), the CVD Prevent Initiative to implement 'upstream' interventions for the prevention of CVD morbidity.[4] Both of these initiatives seek to improve early case finding to prevent avoidable strokes and heart attacks. Both recognise the importance

of supported lifestyle change in conjunction with drug therapies.

Lifestyle or behavioural change requires a degree of individual agency and commitment, which drug therapies do not. Unhealthy lifestyle behaviours are linked to culture and habit, environment, emotions and confidence, which can all moderate an individual's readiness to change and the commitment required to sustain those changes over time.[5] Understanding the attitudes and beliefs that people hold towards diet, exercise and smoking, as well as their perception of their own risk, could assist primary care and public health professionals in providing relevant and effective behavioural advice and social prescribing options. To support evaluations of the NHS Health Check Programme, in 2017, a questionnaire was developed to evaluate patients' awareness of CVD risk at the University College London.[6] This Attitudes and Beliefs about Cardiovascular Disease (ABCD) Risk Questionnaire attempts to provide a short survey drawing from the dominant theoretical models of behaviour change (Trans-Theoretical Model, Health Beliefs Model),[7] covering diet, smoking, exercise and alcohol behaviours, and incorporating a conceptual spread of perceived risk from immediate to lifetime. While a range of validated CVD risk questionnaires exist,[8] and it is common to ask patients to self-report their physical activity, dietary and smoking behaviours through questionnaires and diaries, the ABCD Risk Questionnaire usefully investigates the knowledge, perceptions, beliefs and attitudes that govern these behaviours. The literature suggests that in order to lower measurement errors, larger sample sizes and respondent: item ratios are necessary and that replication is required if the sample size is <300.[9] In the original study, item analysis was carried out on a sample of 110. The necessity to reproduce results was recognised by the authors of the original study:

> Additional studies should be conducted with larger samples to confirm the reliability and validity of the questionnaire. It would be useful to replicate the factor analytic process on an independent, larger sample to confirm the generalisability of these findings.[6]

### Specific objectives
In this study, we revalidated the tool on a sample of the general population in Nottingham to confirm the psychometric properties. Furthermore, we introduced five items related to smoking, which were dropped in the original study due to inadequate case numbers.

To the best of our knowledge, this is the first study that has incorporated items relating to attitudes and intentions towards stopping smoking into the published version of the ABCD Risk Questionnaire and collected sufficient data to submit them to analysis of validity, reliability and factor structure.

In the original ABCD study, over the course of three stages of validity testing (content, face, reliability), items relating to alcohol use and smoking were rejected, leaving

four final subscales: knowledge of CVD risks; perceived risk of heart attack/stroke; perceived benefits and intentions to change; and healthy eating intentions. During exploratory factor analysis (EFA), none of the items relating to alcohol use achieved strong enough loadings to be included in the final scale, and items related to smoking could not be included due to the high proportion of missing data in the experimental sample. The authors of the study note this limitation: 'the questionnaire does not encompass all aspects of CVD risk observed in the general population' and that 'future studies examining populations at increased CVD risk can look into incorporating smoking and alcohol into the ABCD Risk Questionnaire to learn about these individuals' preconceptions and attendance of follow-up care'.[6]

### The present study
Nottingham is one of five global sites of the European Union Horizon 2020 'SPICES'[10] CVD prevention implementation study, which began in 2017. SPICES investigates contextual and health system barriers to the scaling up of successful behaviour change interventions for improved cardiovascular health in low-income, middle-income and high-income European countries. The most recent data (2016) indicate that 'The prevalence of CVD recorded in Nottingham City GP Practices is significantly less than the national (England) average and in comparable areas, despite the CVD mortality rate being significantly higher than average; this partly reflects the differing age structures of the populations, but also indicates significant under-detection/diagnosis'.[11]

The SPICES Nottingham population survey carried out in 2019–2020 used the ABCD Risk Questionnaire alongside the non-clinical INTERHEART CVD risk prediction instrument.[12] The SPICES Study team chose to reintroduce five prewritten items relating to 'intentions and readiness to stop smoking' from the 65-item University College London item pool into the questionnaire due to the high prevalence of smoking in the Nottingham population compared with England averages,[13] and its importance as a CVD risk.[14] This created a 31-item questionnaire. Four items relating to alcohol intake from the same item pool were also considered for inclusion but omitted on two grounds: alcohol-related CVD risk was not a specific focus of the 'SPICES' Study and concerns about the time burden on participants of including the additional items, which can be a barrier to participation.

### METHODS
Incorporating the ABCD Risk Questionnaire into the SPICES Nottingham baseline survey provided cross-sectional study data across a broad sample of adult participants. The dataset generated was therefore suitable for psychometric validation of the original and modified versions of the ABCD Questionnaire. Surveys were administered in person by researchers in the field during attendance at community venues and workplaces.

Administration of the survey took approximately 10 min including provision of consent and confidential communication of results another 10 min on average. Participation was entirely voluntary.

## Participants

Participants were recruited from across the Nottingham conurbation between April 2019 and March 2020 as part of the SPICES Nottingham baseline survey.[10] A purposive sampling method was employed based on community and workplace engagement. This strategy had two components:

1. Engagement of citizens in neighbourhoods through existing community groups, organisations and venues.
2. Engagement of employees in the workplace through large city-based employers.

Community groups were targeted on the basis of the demographic of their membership to ensure that neighbourhoods of differing mean household income, those who are not in employment or of working age, and those from different ethnicities were included. In this way, 327 participants were recruited.

Employers were targeted on the basis of workforce size and policies relating to workforce well-being. Nottingham City Council Adult Care teams and the Rolls-Royce Hucknall site both responded positively and between them provided 156 participants. Nottingham Trent University (NTU) researchers administered the SPICES Nottingham baseline survey individually within the community or workplace setting, and personalised feedback about CVD risks was provided confidentially once the survey had been completed.

Criteria for inclusion included being aged 18+ years, resident in Nottinghamshire, not previously diagnosed with a heart condition, not pregnant and able to provide informed consent.

## Materials

The SPICES baseline survey incorporated the ABCD Risk Questionnaire into a digitised survey instrument created in the Research Electronic Data Capture (REDCap) database system,[15] a secure web application for building and managing online surveys and databases, and the online survey responses were uploaded automatically. No participant data were stored on local devices. Both the ABCD Risk Questionnaire (table 1) and the non-laboratory INTERHEART Questionnaire were included unchanged from their published versions apart from additional five items pertaining to smoking behaviour (table 2).[6]

The surveys were administered in the field by a team of trained researchers recruited from the NTU student body and directly supervised by the SPICES Nottingham coordinator. The surveys were accessed using dedicated tablet computers. Items were reproduced word for word and in the same sequence as the original ABCD Risk Questionnaire with the additional five smoking items inserted after all 26 original items. The five smoking-related items were developed by the authors of the original study through a process of literature review (construct validity), expert panel review (content validity) and modification by focus group (face validity).[6] These five smoking subscale items were included in the 65-item pool developed in the original study but omitted from their analysis due to a high proportion of missing responses.[6]

## Validating the sample

The baseline survey dataset was extracted from REDCap for analysis. Sample was checked for representativeness of the Nottingham population across parameters of age, gender, household income and known rates of physical activity and smoking.

## Data analysis

We took the published 26-item ABCD Risk Questionnaire, introduced five further items relating to smoking behaviours and administered it alongside a validated CVD risk assessment instrument (INTERHEART) to 486 individuals in Nottingham over a period of 12 months. Item, scale and factor reliabilities were remeasured to generate a comparison with the results reported in the original study. Correlation was tested between and among ABCD subscale scores and selected INTERHEART variables, closely matching the methods applied in the original study (online supplemental appendix 3) and results were compared accordingly. After removing incomplete responses, 466 valid cases were entered for analysis, four times the sample size of the original study.

Item and subscale reliabilities were tested using inter-item correlations, corrected item–total correlations and Cronbach's alpha.[16] We performed an EFA to evaluate the dimensionality of items of the original and modified risk scale with and without the smoking items. The EFA was performed using the maximum likelihood extraction and varimax rotation method.[17] Sample and data adequacy was assessed using Kaiser-Meyer-Olkin (KMO) test and Bartlett's test of sphericity was performed to compare an observed correlation matrix with the identity matrix.[18] The adequate number of factors was determined using a scree plot (online supplemental appendix 4). To further test the consistency of factors, we tested using confirmatory factor analysis (CFA). We evaluated the model fit of the CFA using the $X^2$ test, the Tucker-Lewis Index (TLI) and Comparative Fit Index (CFI) and the root mean square error of approximation (RMSEA).[19 20] The analysis was performed using a free statistical software R V.4.0.2. UK postcodes were collected for all participants, which allowed them to be sorted into income deciles using Office for National Statistics (ONS) Index of Multiple Deprivation (IMD) public datasets, allowing correlations to be analysed. Following the methods used in the original study, case data from the 'knowledge' subscale (eight items) were omitted from the analysis since they use a separate response format.[6]

**Table 1** Published ABCD Risk Questionnaire

| Scale | Items |
|---|---|
| Knowledge<br>True/false/don't know<br>Correct score=1<br>Incorrect/don't know score=0<br>Higher sum score=more knowledgeable/more correct about having a heart attack or stroke | 1. One of the main causes of heart attack and stroke is stress |
| | 2. Walking and gardening are considered types of exercise that can lower the risk of having a heart attack or stroke |
| | 3. Moderately intense activity of 2.5 hours a week will reduce your chances of having a heart attack or stroke |
| | 4. People who have diabetes are at higher risk of heart attack or stroke |
| | 5. Managing your stress levels will help you to manage your blood pressure |
| | 6. Drinking high levels of alcohol can increase your cholesterol and triglyceride levels |
| | 7. HDL refers to 'good' cholesterol, and LDL refers to 'bad' cholesterol |
| | 8. A family history of heart disease is not a risk factor for high blood pressure |
| Perceived risk of heart attack or stroke<br>4=strongly disagree, 3=disagree, 2=agree, 1=strongly agree; N/A=0<br>Higher sum score=higher perception of risk of having a heart attack or stroke | 9. I feel I will suffer from a heart attack or stroke some time during my life |
| | 10. It is likely that I will suffer from a heart attack or stroke in the future |
| | 11. It is likely that I will have a heart attack or stroke some time during my life |
| | 12. There is a good chance I will experience a heart attack or stroke in the next 10 years |
| | 13. My chances of suffering from a heart attack or stroke in the next 10 years are great |
| | 14. It is likely I will have a heart attack or stroke because of my past and/or present behaviours |
| | 15. I am not worried that I might have a heart attack or stroke (reverse coded) |
| | 16. I am concerned about the likelihood of having a heart attack or stroke in the near future |
| Perceived benefits and intentions to change<br>4=strongly disagree, 3=disagree, 2=agree, 1=strongly agree; N/A=0<br>Higher average score=higher perceived benefits of diet and exercise and higher perceived readiness for change in regard to exercise and behaviour | 17. I am thinking about exercising at least 2.5 hours a week |
| | 18. I intend or want to exercise at least 2.5 hours a week |
| | 19. When I exercise for at least 2.5 hours a week, I am doing something good for the health of my heart |
| | 20. I am confident that I can maintain a healthy weight by exercising at least 2.5 hours a week |
| | 21. I am not thinking about exercising for 2.5 hours a week (reverse coded) |
| | 22. When I eat five portions of fruits and vegetables a day, I am doing something good for the health of my heart |
| | 23. Increasing my exercise to at least 2.5 hours a week will decrease my chances of having a heart attack or stroke |
| Healthy eating intentions<br>4=strongly disagree, 3=disagree, 2=agree, 1=strongly agree; N/A=0<br>Higher average score=higher perceived readiness for change with regard to healthy dietary behaviour | 24. I am confident that I can eat at least five portions of fruits and vegetables a day within the next 2 months |
| | 25. I am thinking about eating at least five portions of fruits and vegetables a day |
| | 26. I am not thinking about eating at least five portions of fruits and vegetables a day (reverse coded) |

ABCD, Attitudes and Beliefs about Cardiovascular Disease; HDL, high-density lipoprotein; LDL, low-density lipoprotein; N/A, not applicable.

## Patient and public involvement

Patients and/or the public were not involved in the design, or conduct, or reporting, or dissemination plans of this research.

We used the Strengthening the Reporting of Observational Studies in Epidemiology cross-sectional checklist when writing our report.[21]

**Table 2** Additional 'smoking' subscale

| Benefits and intentions to stop smoking | |
|---|---|
| Benefits and intentions to stop smoking<br>4=strongly disagree, 3=disagree, 2=agree, 1=strongly agree; N/A=0<br>Higher average score=higher perceived readiness for change with regard to healthy dietary behaviour | 27. I am thinking of stopping smoking within 2 months |
| | 28. I have reduced or stopped smoking |
| | 29. I intend or want to stop smoking |
| | 30. If I stop smoking, it will reduce my chances of having a heart attack or stroke |
| | 31. I am not thinking about stopping smoking |

N/A, not applicable.

**Parallel Analysis Scree Plots**

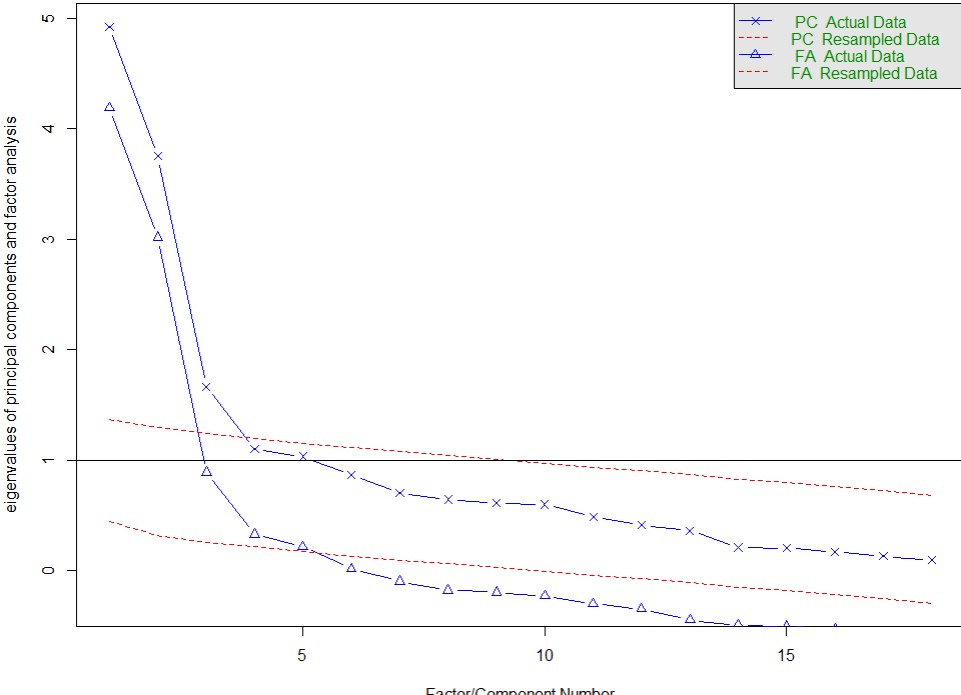

**Figure 1** Scree plot of factor eigenvalues (original published 18 items) Nottingham dataset. FA, factor analysis; PC, principal component.

## RESULTS

### Participants

Participation was voluntary, and self-selection may have been influenced by sensitivities around disclosure of health status and lifestyle habits forming a barrier to those with comorbidities and socially 'questionable' behaviours (heavy smoking, high alcohol intake).

The sample cohort has a 49%:51% gender split, normal distribution of age ranges (18–92) and a distribution of socioeconomic status, which reflects known data about neighbourhood income in Nottingham. Nottingham is the 11th most deprived district in England with higher unemployment, lower education and skills, and shorter life expectancy than the national averages.[22] Using the IMD, a relative measure of deprivation across seven domains, Health and Disability is the domain on which the city's scores are lowest compared with the rest of England. Nevertheless, the mean INTERHEART-predicted risk score for all 466 participants was 10.32, which closely matches the global reported mean for the instrument.[12]

### Smoking subscale

The percentage of smokers in our sample was 15.5%. The proportion of smokers in our sample was therefore higher than the 2019 England average (13.9%) and lower than the Nottingham city population average (20.6%) based on the ONS Annual Population Survey.[23] ONS notes that smoking prevalence estimates by local authority can fluctuate due to smaller sample sizes. Our SPICES Nottingham sample cohort also includes some participants from neighbouring local authorities with different recorded rates of smoking.

The five items in the smoking subscale are measured on the same 4-point response scale as the 18 items submitted for factor analysis in the original published ABCD Risk Questionnaire (strongly agree, agree, disagree, strongly disagree and not applicable).

With the original 18 items, this 'not applicable' response option was not used by any of the SPICES Nottingham study participants. By contrast, within their responses to the items in the 'smoking' subscale, 'not applicable' was the modal answer. Participants chose the 'N/A' response option whenever they reported being a non-smoker. This mirrors the behaviour of the original 110 NHS Health Check attendees who formed the pilot sample cohort for the original study, leaving an insufficient number of smokers in the sample to assess validity and reliability of smoking subscale items. To reduce measurement error in item and factorial analysis, it is recommended overdetermining the ratio of variables to items/factors by using larger sample sizes. No hard rule exists, but at least 10 respondents for each scale item are usually recommended.[24] In the original study, there were insufficient smokers in the sample to achieve this ratio and consequently the smoking subscale items were omitted from the analysis. In the present study, 88 smokers were recorded within the sample and we were therefore able to proceed with item and factorial analysis of the five smoking subscale items.

**Parallel Analysis Scree Plots**

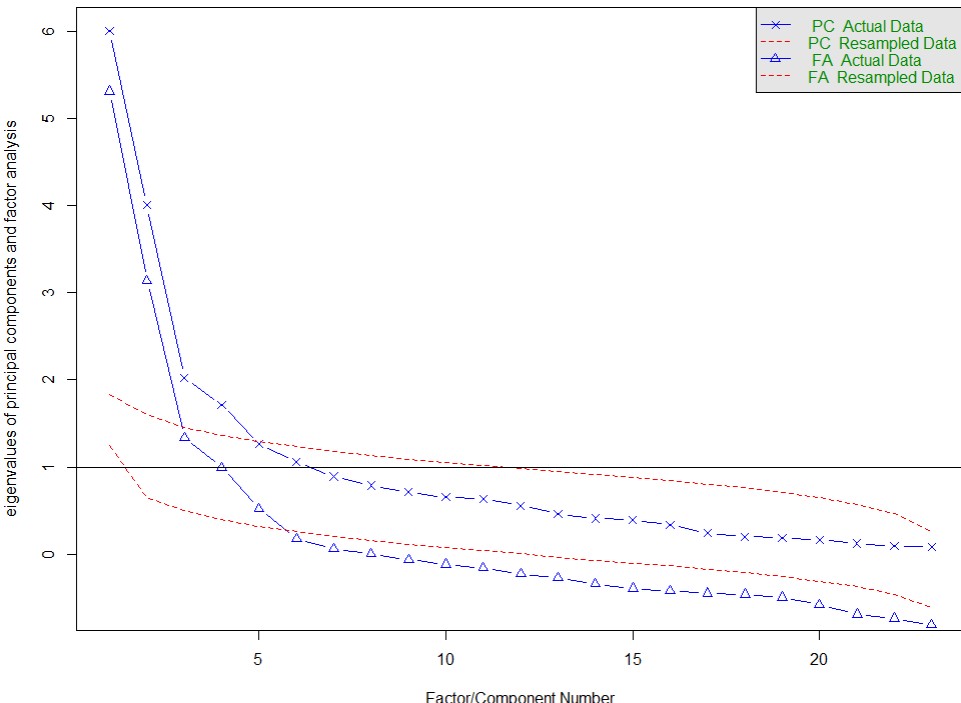

**Figure 2** Scree plot of factor eigenvalues (original published 18 items plus 5 smoking items) Nottingham dataset. FA, factor analysis; PC, principal component.

Subscale alpha values, Cronbach's alpha if item deleted calculated for all items, interitem correlations and corrected item–total correlations were all calculated, mirroring the analysis reported in the original study (online supplemental appendix 5).

Interitem correlations calculated for these five items produced a range between 0.654 and 0.834. All of these five 'smoking' items therefore correlate with one another more strongly than recommended (<0.6) and were considered for rejection. However, we found each item to be qualitatively different, and that the differences were conceptually clear and well expressed in the item wording so that no participant could be expected to confuse one with any other, and they were retained.

Discrimination was confirmed using item–total correlations. These fell between the range 0.751 and 0.906 meaning that all five 'smoking' subscale items are comfortably above the standard cut-off for acceptability of 0.3.

EFA was carried out twice, first with all cases, and then again with 88 confirmed smoking cases. The first operation ensured that factor loadings were not skewed by the lower number of cases reporting smoking behaviours, the second ensured that factor loadings for the remaining subscales where more case data were available were not skewed by outliers.

### Exploratory factor analysis

We conducted EFA on the original 18-item risk perception questionnaire and the modified 23 items (with smoking items). For the original 18 items, a total of 420 observations were included in the analysis, which was sufficient for factor analysis as indicated with KMO of 0.82, which is within the recommended range (0.8–1). The Bartlett's test of sphericity was significant ($X^2$=4235.007, p<0.001) indicating the data are adequate for factor analysis. As a result, a three-factor solution emerged based on the scree plot (figure 1), accounting for 57.4% of the total variance. Factor loading patterns in the present analysis slightly varied from the original subscales. The domains in the original subscales were risk perception, benefit finding and healthy eating intentions. In our analysis, item 14 ('When I eat at least five portions of fruits and vegetables a day, I am doing something good for the health of my heart') showed a better loading to healthy eating intention, which was loaded to benefit finding in the original study (online supplemental appendix 5).

For the modified 23 items (including the smoking subscale), 88 samples were valid and included in the analysis. The KMO was 0.78, which was slightly below the recommended range, but Bartlett's test of sphericity was significant ($X^2$=1223.459, p<0.001), indicating adequacy for factor analysis. The analysis showed that the smoking items loaded to another latent construct resulting in four factors in total (figure 2).

### CFA of the published ABCD Risk Questionnaire

A CFA was undertaken using the SPICES Nottingham dataset to investigate further. Conducting CFA allowed us to construct the subscales of the published ABCD Risk Questionnaire in a three-factor measurement model and test its fit against relevant indices. The original 18-item

**Table 3** CFA fit indices for the original and modified ABCD Questionnaire measurement models

**Original 18-item ABCD**
In the original study of 2017, 18 items were entered into factor analysis. This CFA tests the fit of these original items to their structure using the larger Nottingham SPICES dataset.

| CMIN | P value | CMIN (Chi Square Minimum)/ DF (Degrees of Freedom) | TLI | CFI | RMSEA | RMR |
|---|---|---|---|---|---|---|
| 714.941 | 0.000 | 5.416 | 0.826 | 0.850 | 0.097 | 0.049 |

**Original 18-item ABCD with 5 smoking items added**
In the original study of 2017, items relating to smoking behaviours were developed but could not be included in the published scale due to insufficient data. In the Nottingham SPICES Study, sufficient observations were made to test these smoking items.

| CMIN | P value | CMIN/DF | TLI | CFI | RMSEA | RMR |
|---|---|---|---|---|---|---|
| 994.931 | 0.000 | 4.442 | 0.865 | 0.881 | 0.086 | 0.049 |

**Edited 20-item ABCD with smoking subscale**
As discussed above, independent item analysis and exploratory factor analysis using the independent SPICES Nottingham dataset revealed issues with the continued inclusion of some of the original 'perception of risk' subscale items, and the allocation of an item relating to dietary behaviours in the physical activity behaviours subscale. The published ABCD Questionnaire was edited to remove or reassign the problematic items and retested using CFA.

| CMIN | P value | CMIN/DF | TLI | CFI | RMSEA | RMR |
|---|---|---|---|---|---|---|
| 638.973 | 0.000 | 3.896 | 0.881 | 0.897 | 0.079 | 0.052 |

**Modified 20-item ABCD with smoking subscale**
The measurement model created for the CFA was modified so that items within each ABCD subscale were set to covary with one another.

| CMIN | P value | CMIN/DF | TLI | CFI | RMSEA | RMR |
|---|---|---|---|---|---|---|
| 385.312 | 0.000 | 2.439 | 0.941 | 0.951 | 0.056 | 0.046 |

ABCD, Attitudes and Beliefs about Cardiovascular Disease; CFA, confirmatory factor analysis; CFI, Comparative Fit Index; RMSEA, root mean square error of approximation; TLI, Tucker-Lewis Index.

survey comprising three subscales (perceived risk of heart attack/stroke (eight items); perceived benefits and intentions to change (seven items); healthy eating intentions (three items)) was used to create measurement model in SPSS Amos V.25. The model was then updated to include an additional five-item subscale relating to smoking behaviours.

### Editing the measurement model
The CFA measurement model was then reconstructed removing items which had confused participants and generated high interitem correlations, and additionally reassigning an item relating to dietary behaviour into the dietary behaviour subscale (table 3). This resulted in a four-factor model (perceived risk of heart attack/stroke (six items); perceived benefits and intentions to exercise (six items); healthy eating intentions (four items), perceived benefits and intentions to reduce smoking (five items)). Analysis properties were set to

estimation:maximum likelihood. A scree plot of this amended four-factor version of the questionnaire was also plotted (figure 3).

Similarly, in the 23-item factor analysis, item 14 was loaded to the healthy eating intention. The model fit indices showed a slight improvement as indicated in table 3.

Based on factor loading, interitem correlations and face validity results, we also tested a slightly shorter version of the questionnaire, 20 items, including five smoking items, and the result shows that the model fit improved (CFI=0.941; TLI=0.951; RMSEA=0.056, SRMR (Standardised Root Mean Square Residual)=0.046).

The three published factors achieved a poor fit in CFA (table 3). Including the five smoking-related items, which had performed strongly in EFA as their own latent factor, improved overall model fit slightly but not to an acceptable level.

### Modification of the measurement model
Reviewing modification indices and expected parameter changes for factor loadings and measurement intercepts, we observed an extreme covariance value (116.812) and parameter change (0.209) between two of the risk perception items ('There is a good chance that I will experience a heart attack or stroke in the next 10 years' and 'My chances of suffering a heart attack or stroke in the next 10 years are great'), which had caused confusion for participants in our study.

Removing one of these two items (item 13) and the two other duplicative items (items 9 and 10) from the 'perceived risk of heart attack or stroke' subscale retains the conceptual spread of risk embodied by the items (lifetime, 10 years, near future, behaviour related). Moving the diet-related item (#22) which appears in the 'perceived benefits and intentions to change' over to the 'healthy eating intentions' subscale might allow greater clarity for researchers analysing results from the questionnaire. Covarying items within subscales that generated values above 20 (a high cut-off due to large sample used) resulted in acceptable or good fit across all subscales. Each of the three behaviour-related subscales now contains items drawn from Health Beliefs Model, Trans Theoretical Model and Self Efficacy models providing a sound conceptual basis for comparison. Using EFA to check these results shows the modified subscale structure performs better than the published version (figure 3).

### Other results
Analysing results from ABCD subscales recorded within our sample indicated that mean knowledge of CVD risk factors was 79% and recognition of the benefits of changing behaviour was 85%, but this barely correlated against objectively measured risk (−0.164, sig 0.001, n=436).

**Parallel Analysis Scree Plots**

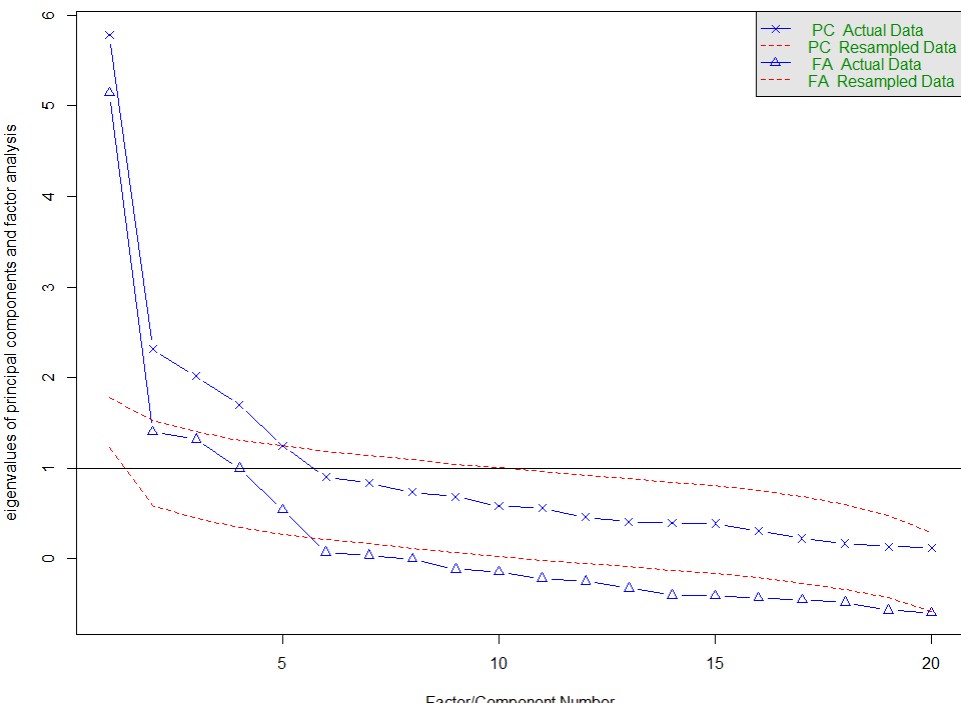

**Figure 3** Scree plot of factor eigenvalues (recommended amended ABCD) Nottingham dataset. ABCD, Attitudes and Beliefs about Cardiovascular Disease; FA, factor analysis; PC, principal component.

## DISCUSSION

Inadequate knowledge and/or a gap between perceived and actual CVD risk in the population could be an obstacle to better health outcomes. Improving an individual's CVD knowledge and risk perception may be important in improving a healthy lifestyle. Measuring CVD knowledge and risk perception may be a method to initiate a healthy lifestyle intervention as well as to monitor and evaluate the impact of interventions. Following this rationale, Woringer and colleagues[6] developed the ABCD Risk Questionnaire in order to measure CVD knowledge and risk perception. In this study, we revalidated the tool on a sample of the general population in Nottingham to confirm the psychometric properties.

The 88 participants in this study who reported smoking is a low number for pilot testing of psychometric scales but it does exceed a 10:1 ratio of cases to variables making it reasonable to proceed to analysis.

Based on EFA and CFA, we confirmed a three-factor structure, which closely matched the results reported in the original study, but differed in certain important respects. Item 14 ('When I eat at least five portions of fruits and vegetables a day, I am doing something good for the health of my heart') showed a better loading to the 'healthy eating intentions' subscale, in contrast to the factor loading in the original study, which placed this item in 'perceived benefits and intentions to change'. This is the only item which loaded onto a different subscale when using the Nottingham dataset; all others continued to load onto their original factors although many of these loaded weakly and failed to meet usual thresholds for validity

(online supplemental appendix 5). The larger number of participants in our dataset (466 compared with 110) provides statistical confidence in the new results, and we therefore modelled this revised allocation of items and factors alongside the original factor allocations in the subsequent CFA. The revised measurement model with item 14 allocated to 'healthy eating intentions' indicated a better fit in CFA results.

These results suggest that the additional five smoking items perform acceptably and should be incorporated into future applications of the ABCD Risk Questionnaire.

## Limitations

Our purposive sampling strategy was non-probabilistic but the resulting sample distribution reflects the population characteristics of Nottingham (online supplemental appendix 6) and therefore permits the generalisation of results to similar urban centres. Because random sampling was not employed, it is not possible to generalise the findings further to a wider population.

Psychometric performance based on reliability calculations and factorial analysis is not an end in itself. The resulting scale has to have some utility in the world and generate results that can add value to existing understanding of beliefs and attitudes to CVD risk. The literature refers to a 'know–do' gap in health education, which is framed as a knowledge translation challenge from research to practice.[25] Analysing results from the ABCD Risk Questionnaire, our findings indicate that this gap also exists within patients/study participants who have recorded high levels of knowledge and motivation

**Table 4** Amended ABCD Risk Questionnaire

| Scale | Items | Coding |
|---|---|---|
| Knowledge | 1. One of the main causes of heart attack and stroke is stress | Correct answers: |
| | 2. Walking and gardening are considered types of exercise that can lower the risk of having a heart attack or stroke | Q1—T |
| | 3. Moderately intense activity of 2.5 hours a week is enough to reduce your chances of having a heart attack or stroke | Q2—T<br>Q3—T |
| | 4. People who have diabetes are at higher risk of having a heart attack or stroke | Q4—T |
| | 5. Managing your stress levels will help you to manage your blood pressure | Q5— T |
| | 6. Drinking high levels of alcohol can increase your cholesterol and triglyceride levels | Q6—T<br>Q7—T |
| | 7. HDL refers to 'good' cholesterol, and LDL refers to 'bad' cholesterol | Q8—F<br>T=True |
| | 8. A family history of heart disease is not a risk factor for high blood pressure | F=False<br>Correct score=1, incorrect or don't know: score=0 |
| Perceived risk of heart attack or stroke | 9. It is likely that I will have a heart attack or stroke some time in my life | 4=strongly disagree, 3=disagree, 2=agree, 1=strongly agree; N/A=0 |
| | 10. There is a good chance I will experience a heart attack or stroke in the next 10 years | 4=strongly disagree, 3=disagree, 2=agree, 1=strongly agree; N/A=0 |
| | 11. It is more likely I will have a heart attack or stroke because of my past and/or present behaviours | 4=strongly disagree, 3=disagree, 2=agree, 1=strongly agree; N/A=0 |
| | 12. I am not worried that I might have a heart attack or stroke | Reverse coded<br>4=strongly disagree, 3=disagree, 2=agree, 1=strongly agree; N/A=0 |
| | 13. I am concerned about the likelihood of having a heart attack or stroke in the near future | 4=strongly disagree, 3=disagree, 2=agree, 1=strongly agree; N/A=0 |
| Perceived benefits and intentions to exercise | 14. I am thinking about exercising at least 2.5 hours a week | 4=strongly disagree, 3=disagree, 2=agree, 1=strongly agree; N/A=0 |
| | 15. I intend or want to exercise at least 2.5 hours a week | 4=strongly disagree, 3=disagree, 2=agree, 1=strongly agree; N/A=0 |
| | 16. When I exercise for at least 2.5 hours a week, I am doing something good for the health of my heart | 4=strongly disagree, 3=disagree, 2=agree, 1=strongly agree; N/A=0 |
| | 17. I am confident that I can maintain a healthy weight by exercising at least 2.5 hours a week | 4=strongly disagree, 3=disagree, 2=agree, 1=strongly agree; N/A=0 |
| | 18. I am not thinking about exercising for 2.5 hours a week | Reverse coded<br>4=strongly disagree, 3=disagree, 2=agree, 1=strongly agree; N/A=0 |
| | 19. Increasing my exercise to at least 2.5 hours a week will decrease my chances of having a heart attack or stroke | 4=strongly disagree, 3=disagree, 2=agree, 1=strongly agree; N/A=0 |
| Perceived benefit and healthy eating intentions | 20. I am confident that I can eat at least five portions of fruits and vegetables a day within the next 2 months | 4=strongly disagree, 3=disagree, 2=agree, 1=strongly agree; N/A=0 |
| | 21. I am thinking about eating at least five portions of fruits and vegetables a day | 4=strongly disagree, 3=disagree, 2=agree, 1=strongly agree; N/A=0 |
| | 22. I am not thinking about eating at least five portions of fruits and vegetables a day | Reverse coded<br>4=strongly disagree, 3=disagree, 2=agree, 1=strongly agree; N/A=0 |
| | 23. When I eat five portions of fruits and vegetables a day, I am doing something good for the health of my heart | 4=strongly disagree, 3=disagree, 2=agree, 1=strongly agree; N/A=0 |
| Benefits and intentions to stop smoking | 24. I am thinking of stopping smoking within 2 months | 4=strongly disagree, 3=disagree, 2=agree, 1=strongly agree; N/A=0 |
| | 25. I have reduced or stopped smoking | 4=strongly disagree, 3=disagree, 2=agree, 1=strongly agree; N/A=0 |
| | 26. I intend or want to stop smoking | 4=strongly disagree, 3=disagree, 2=agree, 1=strongly agree; N/A=0 |
| | 27. If I stop smoking, it will reduce my chances of having a heart attack or stroke | 4=strongly disagree, 3=disagree, 2=agree, 1=strongly agree; N/A=0 |
| | 28. I am not thinking about stopping smoking | Reverse coded<br>4=strongly disagree, 3=disagree, 2=agree, 1=strongly agree; N/A=0 |

ABCD, Attitudes and Beliefs about Cardiovascular Disease; HDL, high-density lipoprotein; LDL, low-density lipoprotein; N/A, not applicable.

to moderate unhealthy behaviours but low levels of success in doing so. This suggests that health education may be failing to stimulate healthy changes in this population, and that other factors (addiction/dependence/social acceptance/lack of resources/time sensitivity) may be limiting the impact of health education even as knowledge of risks and remedies is high. The ABCD Risk Questionnaire enables a careful exploration of the relationships between knowledge, motivation, attitudes and beliefs in relation to CVD risks and their remedies, which may in future be combined with investigation of these confounding factors to improve the effectiveness of future health promotion strategies.

### Other observations

Researchers in the Nottingham SPICES team administering the questionnaire during fieldwork reported that three items within the 'perception of risk of heart attack/stroke' subscale caused consistent difficulties for respondents due to apparent duplication and confusion over fine semantic differences. It was difficult for participants to see a semantic difference between statements 9, 10, 11, 12 and 13, respectively. For items 9, 10 and 11, if we agree that *suffer from* and *have* are synonymous, it is hard to differentiate between *in the future* and *some time during my life* because you would imagine that respondents will be thinking about the future in both cases.

For the questionnaire to be reliable across all sections of the population, including those with limited ability in English (whether native or non-native, first, second or additional language, etc) who may find it particularly hard to differentiate with any confidence between different pairs/sets of statements with largely synonymous meanings, this confusion is a problem. Items 12 and 13 seem to differ mainly only in the possible interpretation of a difference of degree between *good* and *great.*

These face validity issues and their impact can be observed in the interitem correlation results generated during item reliability analysis. In the original study, two items in the perception of risk subscale had been rejected due to correlations in excess of 0.6 leaving eight items. Of these remaining eight items, half had interitem correlations, which exceeded 0.6 when tested against the Nottingham dataset. These were items 9, 10, 11 and 12, which generated interitem correlation values of 0.832, 0.869, 0.616 and 0.729, respectively. Removing items 9, 10 and 13 does not reduce the conceptual range of the 'perception of risk' subscale, which is framed temporally from immediate threat to lifetime risk, it simply removes the duplicate or confusing items. Testing this shortened scale with factor analysis strengthens both item and scale reliability and improves factor loadings (online supplemental appendix 5). We recommend that future versions of the English language ABCD Risk Questionnaire adopt these edits (table 4 and online supplemental appendix 7).

## CONCLUSIONS

The published English language version of the ABCD Risk Questionnaire, with the removal of three problematic 'perception' items, the shift of one item from the 'perceived benefits and intentions to change' subscale into the 'healthy eating intentions' subscale, and the addition of a five-item 'smoking' subscale, performs sufficiently well in validity, reliability and factor analysis with an independent, larger sample to confirm the generalisability of its original published findings. This result supports continued use of the ABCD Risk Questionnaire in the field of CVD prevention research and practice. The inclusion of a smoking behaviours subscale is likely to increase its relevance where smoking behaviours still account for a large proportion of individually modifiable CVD risk in a target population. Although criterion validity has now been established for the 'perception of risk of heart attack/stroke subscale' by two published studies,[6 26] the utility of the remaining subscales individually or in combination has been underexamined. Future studies should investigate the criterion validity of these subscales and the conceptual strength of the items and variables from which they have been composed in order to unambiguously position the resulting survey instrument and evaluate its utility in CVD prevention and treatment practices. Neither this study nor the original published study of 2017 was able to conduct pre/post-intervention measurements in their study design. Measuring using the ABCD survey before an intervention (such as the NHS Health Check) and then again at some time afterwards—in tandem with a validated CVD risk prediction scale (such as INTERHEART or Q Risk 2)—would help to establish the ABCD Risk Questionnaire's sensitivity to change, and perhaps also its ability to discern between types of respondent.

**Acknowledgements** The authors would like to acknowledge the cooperation of Rolls-Royce Hucknall site and Nottingham City Council Adult Care in providing access to employees; Crabtree Farm Community Centre, Middle Street Resource Centre and Self-Help UK in facilitating access to members, users and premises.

**Contributors** Following ICMJE recommendations, MB and HYH assert authorship based on the following four criteria: substantial contributions to the conception or design of the work; or the acquisition, analysis, or interpretation of data for the work; drafting the work or revising it critically for important intellectual content; final approval of the version to be published; and agreement to be accountable for all aspects of the work in ensuring that questions related to the accuracy or integrity of any part of the work are appropriately investigated and resolved. LG and HB assert participating investigator status having served as scientific advisors, critically reviewed the study proposal, and participated in writing or technical editing of the manuscript. LG acts as guarantor.

**Funding** This work was supported by the European Commission Horizon 2020 Non-communicable diseases and the challenge of healthy ageing grant agreement (733356 'SPICES').

**Competing interests** None declared.

**Patient and public involvement** Patients and/or the public were not involved in the design, or conduct, or reporting, or dissemination plans of this research.

**Patient consent for publication** Obtained.

**Ethics approval** This study involves human participants. Ethical approval for the 'SPICES' Nottingham study protocol (incorporating the ABCD Risk Questionnaire) was secured from the Nottingham Trent University College of Business, Law and

Social Sciences on 20 February 2019 (no. 2018/109). Participants gave informed consent to participate in the study before taking part (online supplemental appendix 1).

**Provenance and peer review** Not commissioned; externally peer reviewed.

**Data availability statement** Data are available upon reasonable request. All study data are stored in the NTU controlled data repository with conditions of access set on deposit.

**ORCID iDs**
Mark Bowyer http://orcid.org/0000-0002-1474-5711
Hamid Yimam Hassen http://orcid.org/0000-0001-6485-4193

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
