## [Reviewer comments · BMJ Open]

ARTICLE DETAILS

TITLE (PROVISIONAL)	Psychometric evaluation of the 'Attitudes and Beliefs about Cardiovascular Disease (ABCD) Risk Questionnaire' with validation of a previously untested 'Intentions and Beliefs around Smoking' sub-scale.
AUTHORS	Bowyer, Mark; Hassen, Hamid; Bastiaens, Hilde; Gibson, Linda

VERSION 1 – REVIEW

REVIEWER	Elizabeth L. Ciemins Billings Clin
REVIEW RETURNED	23-Jul-2021

GENERAL COMMENTS	Excellent manuscript, very well written
---

REVIEWER	Y Liu Yangzhou University
REVIEW RETURNED	30-Aug-2021

GENERAL COMMENTS	The study was very interesting. The study evaluated the psychometric and factor properties of an as-yet untested 5 item sub-scale relating to smoking behaviours specifically which based on the Attitudes and Beliefs about Cardiovascular Disease (ABCD) Risk Questionnaire. But I have some doubts, such as 1. Smoking and drinking are all risk factors of CVD, and the authors the study note this limitation 'the questionnaire does not encompass all aspects of CVD risk observed in the general population' and that 'future studies examining populations at increased CVD risk can look into incorporating smoking and alcohol into the ABCD Risk Questionnaire to learn about these individuals' preconceptions and attendance of follow-up care'. But the authors only added the 5 item sub-scale relating to smoking behaviours, and did not add the alcohol related items, why?2. The authors did not tell us how to form the 5 item sub-scale relating to smoking behaviours, what was the basis of 5 item sub-scale?
--

REVIEWER	John Nichols University of Surrey, Nutrition Science
REVIEW RETURNED	07-Oct-2021

GENERAL COMMENTS	Congratulations on completing a difficult project. I found a single
---

	typo on page 9, line 24 where a "s" should, I think, have been an "is". At page 4, lines 52-4: I would include "mental health issues" as an important factor. This is particularly important in relation to smoking and recent evidence suggest that PTSD is a major block to smoking cessation (see references). A weakness of the paper was that the declared aim to validate your questionnaire against demographics of the Nottingham population was done but was sketchy. The data comparing smoking in sample compared with the Nottingham population was given in the discussion section and should, in my opinion, have been reported in the results section but obviously its significance further discussed in the discussion section. References on smoking cessation and PTSD Mathew AR, Cook JW, Japuntich SJ, Leventhal AM (2015). Post-traumatic stress disorder symptoms, underlying affective vulnerabilities, and smoking for affect regulation. The American Journal on Addictions;24: 39–46. Beckham JC, Calhoun PS., Dennis MF, Wilson SM (2012), Dedert EA (2013). Predictors of Lapse in First Week of Smoking Abstinence in PTSD and Non-PTSD Smokers. Nicotine & Tobacco Research; 15:1122–1129. Froeliger B, Beckham JC, Dennis MF, Kozink RV, McClernon FJ (2012). Effects of Nicotine on Emotional Reactivity in PTSD and Non-PTSD Smokers: Results of a Pilot fMRI Study. Advances in Pharmacological Sciences; Volume 2012, Article ID 265724, 6 pages.
--	---

REVIEWER	Chunhua Ma Guangzhou Medical University, School of Nursing Guangzhou Medical University
REVIEW RETURNED	27-Oct-2021

GENERAL COMMENTS	Thank you for authors' significant works. This is an interesting topic, and significant to evaluate the psychometric properties of 'Intentions and Beliefs around Smoking' sub-scale. This is a good research work; I have no further comments for this paper.
--

REVIEWER	Emanuele Fino Nottingham Trent University, psychology
REVIEW RETURNED	03-Nov-2021

GENERAL COMMENTS	Introduction  - It may be useful for a reader to better understand whether any significant aspects relevant to CVD exist at the level of the local (Nottingham) community. For example, are there differences between the prevalence of CVD between the community of interest, and the wider national context? Are there any known differences among local groups, say, at the level of social and demographic characteristics? Please specify. Specific Objectives  - "psychometric and factorial properties" - consider removing 'factorial'. Participants  - Please specify inclusion and exclusion criteria. - How many participants were contacted, in total, and what was the response rate (i.e., 327/x)? - How was the questionnaire administered (e.g., paper and pencil,
--

electronically, etc)? How long did the procedure last? Did participants provide their consent, and how (written, electronic)? Who administered the questionnaires (e.g., research assistant, students, researchers, etc).

Data analysis

- The authors stated that "327 participants were recruited" in previous paragraphs, then they said that "After data cleansing, 466 valid cases were entered for analysis, four times the sample size of the original study." Please could you clarify.
- "After data cleansing" please specify the criteria that guided the data cleansing process (e.g., missing data, unengaged responses, ?)

Results

- Please add a table where you report raw and percentage counts by main demographic and health-related (e.g., smoking) variables, best if column-split by gender.
- Please remove the AMOS settings used. Only leave info about Maximum Likelihood.
- "EFA was carried out twice, firstly with the 88 confirmed smoking cases, and then again with all cases." - why results are presented first on the 420-participant sample, then the 88-participant one?
- Please can you replace "420 samples" with 420 observations, consistently across the paper.
- "Based on factor loading and face validity," - I think the authors should justify more thoroughly their choice of removing items. Item selection should be informed by theory, not factor loadings. Moreover, face validity is not a valid justification.
- I have some concerns about the analysis of the dimensionality of the scale. It is my understanding that the original version of the scale underlies a 5-factor model (knowledge, perceived risk, etc). I would strongly encourage the authors to test the 5-factor model in their CFA as their baseline model, perhaps also including an additional factor including the new 'smoking' items.
- I think the authors could even avoid using EFA, as the main aim of the study is to test a model which seems to be already established. My recommendation is to split the sample of non-smokers in two random sub-samples, then use the first sub-sample to run CFA, and the second to test for invariance vs. smokers. This can be accomplished in AMOS using multigroup analysis.
- As for the reliability, why not using Omega instead of alpha?
- The authors reported to have used scree plots, but those in the Appendix are actually resulting from a methodology called Parallel Analysis, which does more than looking at the explained variance. Could the authors please mention / explain this in their results? However, again, I am not sure this is necessary, as I would recommend sticking to CFA rather than EFA/CFA.

Discussion

- "which is somewhat similar to the original sub-scales." - I think the authors should carefully review this statement and provide some critical comparison.
- Please remove the statement "The 88 participants..." as it is not meaningful information.
- "and we therefore adopted this change in the Confirmatory Factor Analysis" - what do the authors mean?
- I think that the discussion should take into account more the implications of the validated measure for the target population. Most of the currently available statements seem speculative with respect

	to the psychometric properties of the scale. Please try to rewrite the discussion, presenting an overview of results, then discussing the utility and potential impact of the questionnaire. - Please avoid terms like "revalidate". I think the most appropriate way to describe their study is something like "testing the psychometric properties of... in a population of..."
--	---

VERSION 1 – AUTHOR RESPONSE

Reviewer one.

Dr. Elizabeth L. Ciemins, Billings Clin
 Comments to the Author:
 Excellent manuscript, very well written

Response to reviewer:

Many thanks for your kind words.

Reviewer two.

Dr. Y Liu, Yangzhou University
 Comments to the Author:

The study was very interesting. The study evaluated the psychometric and factor properties of an as-yet untested 5 item sub-scale relating to smoking behaviours specifically which based on the Attitudes and Beliefs about Cardiovascular Disease (ABCD) Risk Questionnaire.

But I have some doubts, such as

1. Smoking and drinking are all risk factors of CVD, and the authors the study note this limitation 'the questionnaire does not encompass all aspects of CVD risk observed in the general population' and that 'future studies examining populations at increased CVD risk can look into incorporating smoking and alcohol into the ABCD Risk Questionnaire to learn about these individuals' preconceptions and attendance of follow-up care'. But the authors only added the 5 item sub-scale relating to smoking behaviours, and did not add the alcohol related items, why?

2. The authors did not tell us how to form the 5 item sub-scale relating to smoking behaviours, what was the basis of 5 item sub-scale?

Response to reviewer

1. Excessive alcohol use is a recognised risk factor for cardiovascular diseases, but was not a focus of the parent study (SPICES). The Nottingham SPICES research team discussed the inclusion of four alcohol related items from the original 65 item pool developed as part of the original UCL study. It was finally omitted due to concerns about the growing length of our survey battery (which also included lengthy information and consent items, and the non-clinical INTERHEART instrument). The main body of the article has been updated to clarify this.
2. The five smoking related items were developed by the authors of the original study through a process of literature review (construct validity), expert panel review (content validity), and modification by focus group (face validity). The main body of the article text has been updated to clarify this.

Many thanks for your kind words.

Reviewer three.

Dr. John Nichols, University of Surrey

Comments to the Author:

Congratulations on completing a difficult project. I found a single typo on page 9, line 24 where a "s" should, I think, have been an "is". At page 4, lines 52-4: I would include "mental health issues" as an important factor. This is particularly important in relation to smoking and recent evidence suggest that PTSD is a major block to smoking cessation (see references). A weakness of the paper was that the declared aim to validate your questionnaire against demographics of the Nottingham population was done but was sketchy. The data comparing smoking in sample compared with the Nottingham population was given in the discussion section and should, in my opinion, have been reported in the results section but obviously its significance further discussed in the discussion section.

References on smoking cessation and PTSD

Mathew AR, Cook JW, Japuntich SJ, Leventhal AM (2015). Post-traumatic stress disorder symptoms, underlying affective vulnerabilities, and smoking for affect regulation. *The American Journal on Addictions*;24: 39–46.

Beckham JC, Calhoun PS, Dennis MF, Wilson SM (2012), Dedert EA (2013). Predictors of Lapse in First Week of Smoking Abstinence in PTSD and Non-PTSD Smokers. *Nicotine & Tobacco Research*; 15:1122–1129.

Froeliger B, Beckham JC, Dennis MF, Kozink RV, McClernon FJ (2012). Effects of Nicotine on Emotional Reactivity in PTSD and Non-PTSD Smokers: Results of a Pilot fMRI Study. *Advances in Pharmacological Sciences*; Volume 2012, Article ID 265724, 6 pages.

Response to reviewer

Thank you for your kind words.

Typographic error on page 9 line 24 has been corrected.

"Mental Health Issues" are indeed an important risk factor in cardiovascular disease. So important that both stress and depression are included in the variables measured by the INTERHEART CVD risk questionnaire which was used alongside the ABCD Risk Questionnaire in this study. "Mental Health Issues" were not included in the article text at page 4 lines 52-54 because we are specifically discussing the content and application of the ABCD Risk Questionnaire in this paper so it is important to confine our description to the conceptual framework of the ABCD Risk Questionnaire developed by the team at UCL in 2017 and published in the BMJ that year. Our research question was not 'does the published ABCD Risk Questionnaire include all mediating factors in CVD risk?', but 'are the factorial properties and reliability measurements published in the original ABCD study reproduced when repeated using a larger, independent sample of the UK population?', and of course 'do the five smoking related items omitted from the original published version of the ABCD Risk Questionnaire due to insufficient data perform acceptably when sufficient data is available?'. The references that you have helpfully provided (and which have been noted) are certainly of interest in relation to the expected efficacy of smoking cessation interventions with sufferers of PTSD. We propose that resistance to smoking cessation amongst PTSD sufferers will be reflected in individual scores on this ABCD sub-scale and that future research might look specifically at the strength of this association in the data where a sample of PTSD sufferers could be compared with a control group, but this is not the purpose of our study at this time. We respectfully demur from the change you have suggested.

In the 'Objectives' section of the article Abstract, and in the 'Specific Objectives' section of the introduction to the article we have not made any declaration of intent to 'validate our questionnaire against demographics of the Nottingham population'. Descriptive statistics relating to the demographic make-up of our sample were provided for readers in the first section 'Participants' of the results section for information and context, and to enable judgements to be made by readers about the generalisability of the findings, not because our study sought to compare the sensitivity of the ABCD Risk Questionnaire to differing demographic profiles. We therefore believe that the demographic

information included in the text may be 'sketchy' but is accurate and sufficient given the stated purpose of the paper.

Your observation about the placing of data comparing smoking in our sample with smoking in the Nottingham population is well made, and we have shifted this to the results section as you have recommended.

Reviewer four.

Dr. Chunhua Ma, Guangzhou Medical University

Comments to the Author:

Thank you for authors' significant works. This is an interesting topic, and significant to evaluate the psychometric properties of 'Intentions and Beliefs around Smoking' sub-scale. This is a good research work; I have no further comments for this paper.

Response to reviewer

Thank you for your kind words.

Reviewer five.

Dr. Emanuele Fino, Nottingham Trent University

Comments to the Author:

Reviewer comment

Introduction

- It may be useful for a reader to better understand whether any significant aspects relevant to CVD exist at the level of the local (Nottingham) community. For example, are there differences between the prevalence of CVD between the community of interest, and the wider national context? Are there any known differences among local groups, say, at the level of social and demographic characteristics? Please specify.

Response to reviewer

Introduction. Additional data regarding CVD prevalence in Nottingham city has been included with the appropriate reference as suggested. Further data is available at the referenced resource (<https://www.nottinghaminsight.org.uk/themes/health-and-wellbeing/joint-strategic-needs-assessment/adults/cardiovascular-disease-2016/>) but the observations made about differences between observed CVD outcomes by gender, by age, by postcode, and by ethnicity do not differ in degree or quality from the same differences which can be observed across the UK and in other north European contexts. There is nothing special or different about the patterns of CVD mortality and risk in Nottingham compared more broadly that we should mention here.

Reviewer comment

Specific Objectives

- "psychometric and factorial properties" - consider removing 'factorial'.

Response to reviewer

The word 'factorial' has been removed as recommended

Reviewer comment

Participants

- Please specify inclusion and exclusion criteria.
- How many participants were contacted, in total, and what was the response rate (i.e., 327/x)?
- How was the questionnaire administered (e.g., paper and pencil, electronically, etc)? How long did

the procedure last? Did participants provide their consent, and how (written, electronic)? Who administered the questionnaires (e.g., research assistant, students, researchers, etc).

Response to reviewer

Inclusion and exclusion criteria are now specified as recommended. We have clarified in the text that all surveys were completed face-to-face. In community groups and work-places we attended by invitation and had high rates of take-up. Only those who agreed to participate have been recorded so the response rate is effectively 100%. We set no target on recruitment numbers. Additional details about the way the survey was administered, how long it took, and by whom have all been added to the text (page 6 line 40)

Reviewer comment

Data analysis

- The authors stated that "327 participants were recruited" in previous paragraphs, then they said that "After data cleansing, 466 valid cases were entered for analysis, four times the sample size of the original study." Please could you clarify.
- "After data cleansing" please specify the criteria that guided the data cleansing process (e.g., missing data, unengaged responses?)

Response to reviewer

Previous paragraphs had indicated that 327 participants were recruited (page 7 line 37) at community groups, and 156 (page 7 line 43) through employers, making a total of 486 observations. Criteria for removing observations (incomplete) have now been added (page 7 line 56).

Reviewer comment

Results

- Please add a table where you report raw and percentage counts by main demographic and health-related (e.g., smoking) variables, best if column-split by gender.
- Please remove the AMOS settings used. Only leave info about Maximum Likelihood.
- "EFA was carried out twice, firstly with the 88 confirmed smoking cases, and then again with all cases." - why results are presented first on the 420-participant sample, then the 88-participant one?
- Please can you replace "420 samples" with 420 observations, consistently across the paper.

Response to reviewer

As a validation study, our aim is not to determine the level of risk perception and intention to change and to measure variation across socioeconomic characteristics. In this psychometric analysis, the table on sociodemographic characteristics is included to describe the population of interest to which the tool is applicable. It would certainly be of interest to investigate differences in risk perception and intention to change by age, gender, ethnicity, and socio-economic background but this is beyond the scope of the present study. A follow-up paper is being drafted which further investigates the content, construct and criterion validity of the ABCD Risk Questionnaire drawing on data generated in the Nottingham study and others which have used the ABCD Risk Questionnaire in order to better position the utility of the ABCD Risk Questionnaire in CVD prevention practice.

AMOS settings have been removed as suggested (page 11 lines 56-59).

We have reversed the order of reporting to put the result of EFA against 88 confirmed smoking cases first and the results of EFA against all observations as suggested (page 9 line 27).

The word 'samples' has been replaced with the word 'observations' consistently across the paper as recommended.

Reviewer comment

"Based on factor loading and face validity," - I think the authors should justify more thoroughly their choice of removing items. Item selection should be informed by theory, not factor loadings. Moreover, face validity is not a valid justification.

Response to reviewer

This text has been edited to read 'based on factor loadings, inter-item-correlations, and face validity results...' for greater clarity. In our view, face validity is a necessary step in scale development and where users consistently identify difficulties with the face validity of certain items those items should be reviewed. These difficulties are described in the 'other observations' discussion (page 13 lines 57-60). Inter-item correlations for these same items also indicated problems with the items in that they correlated too highly with one another. We believe that this is an important result and justifies not only having recalculated item analysis results (rather than relying simply on Cronbach's Alpha) with our larger sample, but also our decision to edit the questionnaire and re-test with CFA. Similarly, with our larger independent sample, we were able to observe a significant shift in the factor loading of one item ('When I eat at least 5 portions of fruit and vegetables a day I am doing something good for the health of my heart') which was positioned in the 'Perceived benefits and intentions to change' sub-scale in the original published version, but which sat better in the 'Healthy eating intentions' sub-scale in the results of the Nottingham study EFA. Not only does this new position make more sense conceptually, it generates improved CFA fit results. We believe this is clear in the text.

Reviewer comment

I have some concerns about the analysis of the dimensionality of the scale. It is my understanding that the original version of the scale underlies a 5-factor model (knowledge, perceived risk, etc). I would strongly encourage the authors to test the 5-factor model in their CFA as their baseline model, perhaps also including an additional factor including the new 'smoking' items.

Response to reviewer

The original published version of the scale was constructed using a three-factor model not five (Perceived risk, Perceived benefits and intentions to change, Healthy eating intentions). The Knowledge scale was not entered into factor analysis due to its differing response set and the fact that the 'Knowledge' sub-scale is a criterion-referenced test rather than an investigation of a latent construct. We tested the original three factor model in the CFA reported on page 12 line 12, and again with the addition of the five smoking items on page 12 line 16. Notes have now been included to clarify the labelling of rows in the table.

Reviewer comment

I think the authors could even avoid using EFA, as the main aim of the study is to test a model which seems to be already established.

Response to reviewer

We re-tested EFA because the original published ABCD Risk Questionnaire is *not* yet 'established'. In fact, its authors specifically stated that "it would be useful to replicate factor analytic process on an independent, larger sample to confirm the generalisability of these findings." We therefore prefer to keep this EFA and report it in the paper.

Reviewer comment

My recommendation is to split the sample of non-smokers in two random sub-samples, then use the first sub-sample to run CFA, and the second to test for invariance vs. smokers. This can be accomplished in AMOS using multigroup analysis.

Response to reviewer

Your recommendation to 'split the sample of non-smokers in two random sub-samples, then use the first sub-sample to run CFA, and the second to test for 'invariance vs. smokers' does not seem practical. We think that you are interested in testing for invariance to assess whether the tool can distinguish between groups. If they do not smoke, the observations cannot be used to measure smoking. Test for measurement invariance implies that using the same questionnaire in different groups (smokers vs non-smokers in this case) should measure the same construct in the same way. I think this doesn't work in our case for several reasons.

1. As non-smokers do not have valid response on the scale (all...'NA'), it is not possible to test invariance for smokers vs. non-smokers.
2. The items referring to smoking form a unidimensional sub-scale which is clearly identified by both the EFA and CFA.
3. Even if we choose to split the sample into smokers and non-smokers to test for variance in the way the tool measures against the first three sub-scales (perception of risk, attitudes to physical activity, attitudes to dietary change) we would not expect to observe any variance.

Reviewer comment

As for the reliability, why not using Omega instead of alpha?

Response to reviewer

We used Cronbach's Alpha to report the results of reliability analysis because we needed to be able to make a direct comparison with the reported results of the original study, which had also used Alpha. Some authors believe that coefficient Omega is a more accurate measure of reliability, so in recognition of this we have calculated Omega for each factor and have updated the table to reflect this.

Reviewer comment

The authors reported to have used scree plots, but those in the Appendix are actually resulting from a methodology called Parallel Analysis, which does more than looking at the explained variance. Could the authors please mention / explain this in their results? However, again, I am not sure this is necessary, as I would recommend sticking to CFA rather than EFA/CFA.

Response to reviewer

We understand that Parallel Analysis is a recommended step in evaluating the results of EFA and so is generation of a scree plot. In parallel analysis Eigenvalues are produced from a random data-set using a Monte-Carlo simulation method, and where the Eigenvalues from this random data-set are larger than those generated from the sample dataset the components (in our case) or factors are more likely random noise and can be ignored. A scree plot separately provides a visual indication of which factor loadings (calculated directly from the sample dataset) have an Eigenvalue above 1 (the generally accepted threshold for inclusion). We have used both of these techniques to reach our conclusions. We believe our method is sound and that the reported results are stronger for it. With respect we propose to keep the text as it stands.

We re-tested EFA because the original published ABCD Risk Questionnaire is not yet 'established'. In fact, its authors specifically stated that "it would be useful to replicate factor analytic process on an independent, larger sample to confirm the generalisability of these findings." We therefore prefer to keep this EFA and report it in the paper.

Reviewer comment

Discussion

- "which is somewhat similar to the original sub-scales." - I think the authors should carefully review this statement and provide some critical comparison.
- Please remove the statement "The 88 participants..." as it is not meaningful information.
- "and we therefore adopted this change in the Confirmatory Factor Analysis" - what do the authors mean?

Response to reviewer

We agree that the phrase 'somewhat similar to the original scales' is unscientific. We have edited this statement to read 'which closely matched the results reported in the original study, but differed in certain important respects' (page 13 line 30). The remainder of the paragraph on page 13 describes in detail what the differences are and their potential significance. This paragraph has been further edited for additional clarity and now reads 'The larger numbers of participants in our dataset (466 compared to 110) provides statistical confidence in the new results, and we therefore modelled this revised allocation of items and factors alongside the original factor allocations in the subsequent Confirmatory Factor Analysis. The revised measurement model with item 14 allocated to 'Healthy Eating Intentions' indicated a better fit in CFA results'.

Reviewer comment

I think that the discussion should take into account more the implications of the validated measure for the target population. Most of the currently available statements seem speculative with respect to the psychometric properties of the scale. Please try to rewrite the discussion, presenting an overview of results, then discussing the utility and potential impact of the questionnaire.

Response to reviewer

We believe that psychometric performance based on reliability calculations and factorial analysis is not an end in-itself. The resulting scale has to have some utility in the world, and generate results which can add value to existing understanding of beliefs and attitudes to cardiovascular disease. This is only very lightly touched on in the original paper where it states that 'the questionnaire can be used to assess patients' understanding of CVD risk'. We believe that because there is a recognised gap between 'knowing' and 'doing' in relation to CVD risk factors which means that much health education may be failing to stimulate the healthy changes that we would like to see in the population, it is very important to consider the attitudes and beliefs about elective change in relation to risky lifestyle behaviours which may be mediating this relationship. It does not seem to be enough simply to educate vulnerable people to the nature of the risks in order to stimulate the necessary changes. Although socio-economic factors will also play a part here, and there may be additional psychological factors (such as 'present-bias') which also mediate this space, the ABCD Risk Questionnaire goes a long way to investigating and measuring the personal beliefs and attitudes which are in operation here. The conclusion has been updated to make this clear.

Nevertheless, neither this paper or the original published paper of 2017 was able to conduct pre-post measurements in their study design. Measuring using the ABCD survey before an intervention (such as the NHS Health Check) and then again at some time afterwards- in tandem with a validated CVD risk prediction scale (such as INTERHEART or Q Risk 2) would help to establish the ABCD Risk Questionnaire's sensitivity to change, and perhaps also its ability to discern between types of

respondent. We have updated the strengths and weaknesses statement to reflect this. A separate paper is being drafted which reports our detailed analysis of content, construct, and criterion validity in the ABCD Risk Questionnaire which we hope will enable researchers and clinicians to better position the use of the ABCD Risk Questionnaire in professional practice.

Reviewer comment

Please avoid terms like "revalidate". I think the most appropriate way to describe their study is something like "testing the psychometric properties of... in a population of..."

Response to reviewer

Text corrected as recommended. Many thanks for your thoughtful observations.

VERSION 2 – REVIEW

REVIEWER	Y Liu Yangzhou University
REVIEW RETURNED	03-Apr-2022

GENERAL COMMENTS	The study was very interesting and beneficial, and provided a very good measuring tool to assist public health practitioners and researchers to survey patient or public intentions and beliefs around three key areas of individually modifiable risk (Physical Activity, Diet, Smoking). The study provided evidence of validity, reliability and generalisability of results obtained using the Attitudes and Beliefs about Cardiovascular Disease (ABCD) Risk Questionnaire with a sample of the English population surveyed within the ‘SPICES’ Horizon 2020 project (Nottingham study site), and to specifically evaluate the psychometric and factor properties of an as-yet untested 5 item sub-scale relating to smoking behaviours. But I have some questions below:  1. Sampling Method: The author did not introduce the sampling method to us in detail, how the study participants were extracted? Whether there was bias? 2. The people who come from community would include older adults, how did you obtain the questionnaire answers? And the basic information of the population should be described by table, we should know the suitable crowd of the questionnaire.
--

REVIEWER	John Nichols University of Surrey, Nutrition Science
REVIEW RETURNED	13-Jan-2022

GENERAL COMMENTS	You have developed a useful tool for preventive projects and research. The statistical details are a bit difficult to follow and an explanation for the purpose of statistical tests could be more complete. For instance, page 11 lines 18-20 was difficult to understand and could be explained better using language other than statistical jargon. I found a typo on page 20, line 22 where the word "the" should have been "this". I expect it was corrected before it was presented to the general public? In future research, you might find it helpful to look at the balance between motivators for behavioral change (i.e. stop smoking) and de-motivators. I have been trying to summarize this in relation to smoking cessation and I am attaching my finding as this data may be relevant to your future research. The importance of recognizing
--

	and treating PTSD in smokers before attempting smoking cessation is an example.
--	---

REVIEWER	Rosemary Hiscock University of Bath
REVIEW RETURNED	29-Jun-2022

GENERAL COMMENTS	This study conducts factor analysis on a new dataset which enables the authors to reduce the number of items when modifying a questionnaire. The new dataset allowed the authors to include items on smoking in their factor analysis as there were adequate numbers of smokers surveyed (which was not the case in the original dataset). The questionnaire is about understanding personal risk of heart disease. In the introduction bring out:  1) Why the ABCD is needed -there are lots of existing measures of smoking, activity and diet and surely CVD risk 2) The importance of replicating findings given problems of replication when it has been done in previously https://www.news-medical.net/life-sciences/What-is-the-Replication-Crisis.aspx I have tried to use the page numbers on the bottom right of the screen except for the appendix where they were not available. Line numbers I have tried to refer to the number closest to the line P2 line 1-4 The title needs to change because you mostly retest and modify the original questionnaire – adding the smoking subscale is only a small part of your work P2 line 14 I would call your sample workplace in addition to community Line 31 -32 I think you should be more precise e.g. “CVD is also the biggest morbidity/disease?? contributor to the inequalities in Healthy Life Expectancy” because one would expect a health behaviour rather than a disease in the sentence P9 most of lines 5 to 7 should be around line 24 on page 6 P9 Line 7-8 what do you mean by remeasured – compared to the original validation study or was it measured twice for your own participants? P9 Line 24 what do you mean by ‘case data’ do you mean the knowledge subscale was excluded from the analysis? Was this consistent with the original validation? P9 Line 34 I’m not sure a sample can be ‘strongly parametric’ only a variable – what do you mean by this term? P9 Line 39 ‘does worst’ is a value just meant perhaps “scores are more markedly lower than elsewhere” P9 Line 41 what about for the UK or England? P10 line 2 In 2019 13.9% adults in England smoked https://www.ons.gov.uk/peoplepopulationandcommunity/healthandsocialcare/healthandlifeexpectancies/bulletins/adultsmokinghabitsingreatbritain/2019 P11 I can’t read the notes because the table is on top P11 Table 3 please define all the abbreviations on this table. Table 3 seems much too technical to be in the main text. Instead I would put the modified questionnaire in the appendix p49 of 71 P13 line 32 Put this in a limitations section and add something on the following: There is increasing evidence that even though people understand
---

	that certain foods, especially ultra processed foods, are unhealthy they are unable to stop eating them as they are addictive. This limits the usefulness of education (similar issues with smoking dependence). Therefore research should increasingly be looking at the supply side rather than the demand side https://www.mdpi.com/2072-6643/14/1/23 https://www.ncbi.nlm.nih.gov/pmc/articles/PMC7694501/ https://medium.com/myselfies/ultra-processed-food-is-messing-with-your-brain-2da37c98a09e Page 14 line 37 you need to add “an intervention” after “pre-post” Appendix 5 You have not explained in your methods what the simulated and resampled data are and what the lines on the graph mean as far as I can see. Also it would be better that the X axis has a scale where all numbers are included rather than at intervals at five Any tables included in the appendix must have the full item rather than just the variable name to help the reader Note you have currently still got notes to yourselves in the appendix on whether or not things should be included
--	--

VERSION 2 – AUTHOR RESPONSE

Reviewer: 3

Dr. John Nichols, University of Surrey

Comments to the Author:

You have developed a useful tool for preventive projects and research. The statistical details are a bit difficult to follow and an explanation for the purpose of statistical tests could be more complete. For instance, page 11 lines 18-20 was difficult to understand and could be explained better using language other than statistical jargon.

I found a typo on page 20, line 22 where the word "the" should have been "this". I expect it was corrected before it was presented to the general public?

Response: page and line numbers do not correspond with manuscripts held by the authors. Authors will consider and edit these issues when this has been resolved.

In future research, you might find it helpful to look at the balance between motivators for behavioural change (i.e. stop smoking) and de-motivators. I have been trying to summarize this in relation to smoking cessation and I am attaching my finding as this data may be relevant to your future research. The importance of recognizing and treating PTSD in smokers before attempting smoking cessation is an example.

Response: Thank you for your paper and we will certainly consider this in our future work.

Reviewer: 2

Dr. Y Liu, Yangzhou University

Comments to the Author:

The study was very interesting and beneficial, and provided a very good measuring tool to assist public health practitioners and researchers to survey patient or public intentions and beliefs around three key areas of individually modifiable risk (Physical Activity, Diet, Smoking). The study provided evidence of validity, reliability and generalisability of results obtained using the Attitudes and Beliefs about Cardiovascular Disease (ABCD) Risk Questionnaire with a sample of the English population surveyed within the ‘SPICES’ Horizon 2020 project (Nottingham study site), and to specifically evaluate the psychometric and factor properties of an as-yet untested 5 item sub-scale relating to smoking behaviours.

But I have some questions below:

1. Sampling Method: The author did not introduce the sampling method to us in detail, how the study participants were extracted? Whether there was bias?

Response: Details of sampling methods used described in Methods section (page 6 line 29 > page 7 line 4). A purposive rather than a randomised sampling method was used (as stated), but this would not affect the reliability of the psychometric testing results. Purposive sampling may, however, affect the generalisability of the results although the resulting sample is strongly reflective of the target population in terms of gender split, age range, and socio-economic status as discussed in the results section page 9 lines 37-42.

Potential source of sampling bias noted in Results page 9 lines 34-36.

2. The people who come from community would include older adults, how did you obtain the questionnaire answers? And the basic information of the population should be described by table, we should know the suitable crowd of the questionnaire.

Response: Table describing population characteristics now included in appendices (Appendix 8)

Reviewer: 6

Dr. Rosemary Hiscock, University of Bath

Comments to the Author:

This study conducts factor analysis on a new dataset which enables the authors to reduce the number of items when modifying a questionnaire. The new dataset allowed the authors to include items on smoking in their factor analysis as there were adequate numbers of smokers surveyed (which was not the case in the original dataset). The questionnaire is about understanding personal risk of heart disease.

In the introduction bring out:

1) Why the ABCD is needed -there are lots of existing measures of smoking, activity and diet and surely CVD risk

Response: Why the ABCD is needed now described in Introduction page 5 lines 15-19.

1.

2) The importance of replicating findings given problems of replication when it has been done in previously <https://eur03.safelinks.protection.outlook.com/?url=https%3A%2F%2Fwww.news-medical.net%2Flife-sciences%2Fwhat-is-the-Replication-Crisis.aspx&data=05%7C01%7Cmark.bowyer%40ntu.ac.uk%7C580ed3147c604d1588ac08da60553528%7C8acbc2c5c8ed42c78169ba438a0dbe2f%7C1%7C0%7C637928216338759483%7CUnknown%7CTWFpbGZsb3d8eyJWljojMC4wLjAwMDAiLCJQIjoiV2luMzIiLCJBTiI6Ikl1haWwiLCJXVCi6Mn0%3D%7C3000%7C%7C%7C&odata=ovRYZ8oDxzuySCEXWgmKZ%2FTj%2FdNyUrD8KUWmpuWEYfY%3D&reserved=0>

Response: Importance of replication of findings now described in Introduction page 5 line 19.

I have tried to use the page numbers on the bottom right of the screen except for the appendix where they were not available. Line numbers I have tried to refer to the number closest to the line

P2 line 1-4 The title needs to change because you mostly retest and modify the original questionnaire – adding the smoking subscale is only a small part of your work

Response: Title now changed to 'Psychometric evaluation of the 'Attitudes and Beliefs about Cardiovascular Disease (ABCD) Risk Questionnaire' with validation of a previously untested 'Intentions and Beliefs around Smoking' sub-scale' to reflect the full scope of the work.

P2 line 14 I would call your sample workplace in addition to community

Response: 'Workplace sample' added alongside 'community sample' page 2 line 14.

Line 31 -32 I think you should be more precise e.g. "CVD is also the biggest morbidity/disease?? contributor to the inequalities in Healthy Life Expectancy" because one would expect a health behaviour rather than a disease in the sentence

Response: 'CVD morbidity is also the biggest contributor to the inequalities in healthy life expectancy between members of the wealthiest neighbourhoods and the most deprived.[2]' retained because the reference explains the link between CVD morbidity and QALYs and DALYs. The authors note that many academic papers relating to CVD describe the burden on society and the health services in terms of mortality, whereas for health service policy makers and commissioners, it is the early onset of CVD morbidity with its consequent management and treatment costs which are of pressing concern. In addition productivity lost to the economy of long term disability and impact on individual and household well-being and quality of life is associated with rising CVD morbidity rather than mortality. We wanted to ensure that these impacts of CVD were referenced in our introduction as the backdrop to our interest in the subject and with word-count restrictions we felt this sentence and its associated reference encapsulated the issue adequately.

P9 most of lines 5 to 7 should be around line 24 on page 6

Response: Lines edited and moved as suggested.

P9 Line 7-8 what do you mean by remeasured – compared to the original validation study or was it measured twice for your own participants?

Response: Text changed to 'computed to generate a comparison to the results reported in the original study' for clarity page 9 line 5.

P9 Line 24 what do you mean by 'case data' do you mean the knowledge subscale was excluded from the analysis? Was this consistent with the original validation?

Response: Phrase 'following the methods used in the original study' inserted at page 9 line 22. 'Case data' refers to all recorded responses by study participants against these variables.

P9 Line 34 I'm not sure a sample can be 'strongly parametric' only a variable – what do you mean by this term?

Response: Phrase 'strongly parametric' removed.

P9 Line 39 'does worst' is a value just meant perhaps "scores are more markedly lower than elsewhere"

Response: Phrase 'Does worst' changed to 'city's scores are lowest'.

P9 Line 41 what about for the UK or England?

Response: page 9 line 34 provides the England context.

P10 line 2 In 2019 13.9% adults in England

smoked <https://eur03.safelinks.protection.outlook.com/?url=https%3A%2F%2Fwww.ons.gov.uk%2Fpeoplepopulationandcommunity%2Fhealthandsocialcare%2Fhealthandlifeexpectancies%2Fbulletins%2Fadultsmokinghabitsingreatbritain%2F2019&data=05%7C01%7Cmark.bowyer%40ntu.ac.uk%7C580ed3147c604d1588ac08da60553528%7C8acbc2c5c8ed42c78169ba438a0dbe2f%7C1%7C0%7C637928216338759483%7CUnknown%7CTWFpbGZsb3d8eyJWljoimC4wLjAwMDAiLCJQljoiv2luMzliLCJBTiil6lk1haWwiLCJXVCI6Mn0%3D%7C3000%7C%7C%7C&sd=07yig%2FDk04Wi3o5kdUvuYzcS6%2FYdjm1DqPnwxP9gWio%3D&reserved=0>

Response: Text changed to include more recent UK smoking data for comparison.

P11 I can't read the notes because the table is on top

Response: Notes now incorporated into main text for greater clarity and legibility.

P11 Table 3 please define all the abbreviations on this table. Table 3 seems much too technical to be in the main text. Instead I would put the modified questionnaire in the appendix p49 of 71

Response: Abbreviations now defined on pages 11-12 under new section 'selection of fit indices'. Table 3 retained in main text due to importance of results for later conclusions.

P13 line 32 Put this in a limitations section and add something on the following:

There is increasing evidence that even though people understand that certain foods, especially ultra processed foods, are unhealthy they are unable to stop eating them as they are addictive. This limits the usefulness of education (similar issues with smoking dependence). Therefore research should increasingly be looking at the supply side rather than the demand side

<https://eur03.safelinks.protection.outlook.com/?url=https%3A%2F%2Fwww.mdpi.com%2F2072-6643%2F14%2F1%2F23&data=05%7C01%7Cmark.bowyer%40ntu.ac.uk%7C580ed3147c604d1588ac08da60553528%7C8acbc2c5c8ed42c78169ba438a0dbe2f%7C1%7C0%7C637928216338759483%7CUnknown%7CTWFpbGZsb3d8eyJWljoimC4wLjAwMDAiLCJQljoiv2luMzliLCJBTiil6lk1haWwiLCJXVCI6Mn0%3D%7C3000%7C%7C%7C&sd=CbEMMmj8ToKkBBNvr0jG2HD3K%2FPPDCPX%2Bm%2FdbNn0Tlo%3D&reserved=0>

<https://eur03.safelinks.protection.outlook.com/?url=https%3A%2F%2Fwww.ncbi.nlm.nih.gov%2Fpmc%2Farticles%2FPMC7694501%2F&data=05%7C01%7Cmark.bowyer%40ntu.ac.uk%7C580ed3147c604d1588ac08da60553528%7C8acbc2c5c8ed42c78169ba438a0dbe2f%7C1%7C0%7C637928216338759483%7CUnknown%7CTWFpbGZsb3d8eyJWljoimC4wLjAwMDAiLCJQljoiv2luMzliLCJBTiil6lk1haWwiLCJXVCI6Mn0%3D%7C3000%7C%7C%7C&sd=yKHGPzd7QyxX9yqDuZmjZ1w63I%2FIF1%2BDGdZAFck2rhl%3D&reserved=0>

<https://eur03.safelinks.protection.outlook.com/?url=https%3A%2F%2Fmedium.com%2Fmyselfies%2Fultra-processed-food-is-messing-with-your-brain-2da37c98a09e&data=05%7C01%7Cmark.bowyer%40ntu.ac.uk%7C580ed3147c604d1588ac08da60553528%7C8acbc2c5c8ed42c78169ba438a0dbe2f%7C1%7C0%7C637928216338759483%7CUnknown%7CTWFpbGZsb3d8eyJWljoimC4wLjAwMDAiLCJQljoiv2luMzliLCJBTiil6lk1haWwiLCJXVCI6Mn0%3D%7C3000%7C%7C%7C&sd=wqt%2Fjcdq8HrVFLPGJHtoYu4PyKeHT%2FH%2F0>

Response: Assertions about the addictiveness of ultra-processed foods and other supply-side factors are outside the scope of this paper. In fact, 'supply-side' issues are of great interest to the authors inasmuch as they limit or interrupt the scope of personal agency in healthy behaviour change. The ABCD Risk questionnaire deals only with demand-side issues and as such is a useful and necessary tool to begin to quantify the 'demand-side profile' of potentially vulnerable citizens/patients (vulnerable to CVD). Results should however be interpreted carefully since barriers to successful behaviour change may be both internal (psychological affects such as 'present bias' or fatalism) or external (socio-economic effects like the cost of healthy food, gym membership, or access to fresh vegetables in poorer neighbourhoods). Some of these issues were also explored in the 'SPICES' study, and may be discussed in future publications.

Page 14 line 37 you need to add "an intervention" after "pre-post"

Response: 'Intervention' added in main text.

Appendix 5 You have not explained in your methods what the simulated and resampled data are and what the lines on the graph mean as far as I can see. Also it would be better that the X axis has a scale where all numbers are included rather than at intervals at five

Response: None of the data was simulated. Scree plot 'elbow graphs' are standard in EFA reporting. Whilst it would be possible to reproduce these scree plots with all axis metric numbers included we feel that this would make the figures untidy and less legible rather than more legible (we tried it).

Any tables included in the appendix must have the full item rather than just the variable name to help the reader

Response: Full item rather than variable names now used.

Note you have currently still got notes to yourselves in the appendix on whether or not things should be included

Response: 'Notes to self' in Appendix 5 now removed thank you.

VERSION 3 – REVIEW

REVIEWER	Rosemary Hiscock University of Bath
REVIEW RETURNED	12-Aug-2022

GENERAL COMMENTS	This paper replicates validation of a questionnaire on understanding of causes of cardiovascular disease and for the first time evaluates a subscale of the questionnaire on smoking. Some of my concerns have been adequately addressed. I have the following further comments: P5 line 19 I was looking here for a reference on how replicating studies is useful in general e.g. https://www.nature.com/articles/533452a P8 line 7 reference for original study
--

	P9 line 23 reference for original study P9 line 37 city scores are lowest compared to the rest of England compared to the other subscales? P11 line 22 do you mean an insufficient number/proportion of smokers in the sample? P13 line 6 I still think you need to include the revised version of the questionnaire in the main text – if you are recommending researchers in future should use an amended version its important to be clear what they should be using. People often don't look at supplemental files Discussion Add a limitations section with comments on: (1) whether the sample were sufficiently representative to be able to generalise to the population of Nottingham or England etc. you don't have a random sample which is a concern (2) limitations of usefulness of education in improving public health e.g addiction/dependence/social acceptance prevents people being able to act even if they know behaviour is bad for their health Appendix 5 figures 1,2 &3 You say there was no simulated data in your response. Please either include an explanation of why these elbow graphs include simulated and resampled lines (the red lines)– or remove the red lines
--	--

VERSION 3 – AUTHOR RESPONSE

Reviewer Comments

Dr. Rosemary Hiscock, University of Bath

This paper replicates validation of a questionnaire on understanding of causes of cardiovascular disease and for the first time evaluates a subscale of the questionnaire on smoking.

Reviewer comment:

Some of my concerns have been adequately addressed. I have the following further comments:
P5 line 19 I was looking here for a reference on how replicating studies is useful in general e.g. <https://www.nature.com/articles/533452a>

Author response:

We recognise the importance and challenge of replication of results in medical research. This is especially challenging where study protocols (such as ours) are not conducted using strict laboratory or other experimental conditions. This study did not attempt to replicate the whole psychometric development process employed in the original study. The original study developed an item pool of 85 items which were drawn from a literature review then reduced to 65 through content and face validity testing, and then reduced again to 26 items in reliability testing (item and factor analysis). Our study sought to retest these 26 items using a larger independent dataset as requested by the authors of the original study. The fact that our re-testing of the reliability of the questionnaire by item and by subscale has highlighted some issues with the original published results, and that the new data allowed us to recommend an amended version shows how important and useful our limited replication has been.

Because we have not attempted a 'complete' replication of the original study protocol, and because this was not a medical trial which could be fully replicated and this was not our primary aim, in the

interests of word-count, we have kept the revised wording already submitted for review, and provided a reference to the replication challenge article instead. We hope you feel this is sufficient in the circumstances.

Reviewer comment:

P8 line 7 reference for original study

Author response:

Reference to original study now included.

Reviewer comment:

P9 line 23 reference for original study

Author response:

Reference to original study now included.

Reviewer comment:

P9 line 37 city scores are lowest compared to the rest of England compared to the other subscales?

Author response:

Text edited to clarify (compared to rest of England).

Reviewer comment:

P11 line 22 do you mean an insufficient number/proportion of smokers in the sample?

Author response:

Text edited to clarify (proportion)

Reviewer comment:

P13 line 6 I still think you need to include the revised version of the questionnaire in the main text – if you are recommending researchers in future should use an amended version its important to be clear what they should be using. People often don't look at supplemental files

Author response:

We appreciate the point you make. The original published version of the questionnaire is included in the manuscript as Table 1 because we make references to sub-scales and item numbers in the methods and results sections so the reader needs to be able to cross-reference. We have two options for including the amended version in the main text. We could include it as an additional table (Table 4) at the end of the results section, or we could somehow combine both the original and amended versions with appropriate labelling. We are not sure what would be best and will ask for advice from the BMJ editors. The original published version included the final version of the questionnaire in the appendices only.

Reviewer comment:

Discussion

Add a limitations section with comments on:

(1) whether the sample were sufficiently representative to be able to generalise to the population of Nottingham or England etc. you don't have a random sample which is a concern

Author response:

An additional paragraph about sampling strategy has been inserted into the discussion section.

Reviewer comment:

(2) limitations of usefulness of education in improving public health e.g addiction/dependence/social acceptance prevents people being able to act even if they know behaviour is bad for their health

Author response:

Text of second paragraph in the 'discussion-limitations' section has been edited to reflect this observation.

Reviewer comment:

Appendix 5 figures 1,2 &3 You say there was no simulated data in your response. Please either include an explanation of why these elbow graphs include simulated and resampled lines (the red lines)– or remove the red lines

Author response:

We removed the red lines referring to simulated data in all the scree plots. However, due to the default output from the analysis, red lines referring to resampled data are still there. Having the red lines is more informative to look at how the eigenvalues drop steeply as compared to the resampled data (resampling from our data). The principle is to retain, at maximum, the number of factors with observed eigenvalues that are larger than those extracted from corresponding factors based on resampled/noise data. Thus, it's reasonable to keep the resampled data.

VERSION 4 – REVIEW

REVIEWER	Rosemary Hiscock University of Bath
REVIEW RETURNED	22-Sep-2022

GENERAL COMMENTS	This study looks at the properties of a scale measuring CVD risk in a new sample In the introduction the authors say they are confirming "the reliability and generalisability of the ABCD risk questionnaire" because but in their response to my review they say they are not doing a full replications so they should not draw attention to the usefulness of replication in general. If this replication is a useful addition to the literature the authors need to talk about replication in general to justify this major part of their analysis p10 line 8 -its the proportion not the number of smokers p10 line 22 -did the analysis require a certain proportion of smokers or an absolute number of smokers (which was not met originally) - given the previous error in line 8, I'm just checking 'proportion' is correct here Scree plots - if you are unable to amend the graph to remove the red lines you need to explain either in the methods, in the appendix or in a footnote under the graph what they refer to
--

	If the editor thinks it is too much to have both the original and the amended versions of the scale in the text, I would have the amended version and either refer to the original version with its original reference or put the original version in a supplementary file if it has not been published before. Alternatively include a table with both versions highlighting the changes p16 line 16. You have not explicitly addressed that the sample was not random
--	---